# Predictive Value of Carcinoembryonic Antigen in Symptomatic Patients without Colorectal Cancer: A Post-Hoc Analysis within the COLONPREDICT Cohort

**DOI:** 10.3390/diagnostics10121036

**Published:** 2020-12-02

**Authors:** Noel Pin-Vieito, María José Iglesias, David Remedios, Victoria Álvarez-Sánchez, Fernando Fernández-Bañares, Jaume Boadas, Eva Martínez-Bauer, Rafael Campo, Luis Bujanda, Ángel Ferrández, Virginia Piñol, Daniel Rodríguez-Alcalde, Martín Menéndez-Rodríguez, Natalia García-Morales, Cristina Pérez-Mosquera, Joaquín Cubiella

**Affiliations:** 1Gastroenterology Department, Complexo Hospitalario Universitario de Ourense, Centro de Investigación Biomédica en Red de Enfermedades Hepáticas y Digestivas (CIBERehd), 32005 Ourense, Spain; Maria.Jose.Iglesias.Varela@sergas.es (M.J.I.); David.Rafael.Remedios.Espino@sergas.es (D.R.); Cristina.Perez.Mosquera@sergas.es (C.P.-M.); Joaquin.Cubiella.Fernandez@sergas.es (J.C.); 2Instituto de Investigación Biomedica Galicia Sur, 32005 Ourense, Spain; 3Department of Biochemistry, Genetics and Immunology, Faculty of Biology, University of Vigo, 36200 Vigo, Spain; 4Gastroenterology Department, Complejo Hospitalario de Pontevedra, 36001 Pontevedra, Spain; victoria.alvarez.sanchez@sergas.es; 5Gastroenterology Department, Hospital Universitari Mútua de Terrassa, CIBERehd, 08221 Terrassa, Spain; ffbanares@mutuaterrassa.es; 6Gastroenterology Department, ConsorciSanitari de Terrassa, 08221 Terrassa, Spain; jboadas@cst.cat; 7Gastroenterology Department, Hospital de Sabadell, Corporació Sanitàriai Universitària Parc Taulí, 08208 Sabadell, Spain; emartinezb@tauli.cat (E.M.-B.); rcampo@tauli.cat (R.C.); 8Donostia Hospital, Biodonostia Institute, University of the Basque Country UPV/EHU, CIBERehd, 20010 San Sebastian, Spain; luis.bujandafernandezdepierola@osakidetza.eus; 9Servicio de Aparato Digestivo, Hospital Clínico Universitario, IIS Aragón, University of Zaragoza, CIBERehd, 50009 Zaragoza, Spain; angel.ferrandez@telefonica.net; 10Gastroenterology Department, Hospital Dr. Josep Trueta, 17007 Girona, Spain; vpinol.girona.ics@gencat.cat; 11Digestive Disease Section, Hospital Universitario de Móstoles, 28935 Madrid, Spain; drodrigueza@salud.madrid.org; 12Lavadores Primary Care Health Centre, Área de Saúde de Vigo, Pontevedra, 36214 Vigo, Spain; martinmenendezrodriguez@hotmail.com; 13Gastroenterology Department, Complexo Hospitalario Universitario Vigo, Pontevedra, 36001 Vigo, Spain; Natalia.Garcia.Morales@sergas.es

**Keywords:** biochemical diagnosis, carcinoembryonic antigen, colonoscopy, early cancer, gastrointestinal cancer, symptoms, tumor biomarkers

## Abstract

We aimed to assess the risk of cancer in patients with abdominal symptoms after a complete colonoscopy without colorectal cancer (CRC), according to the carcinoembryonic antigen (CEA) concentration, as well as its diagnostic accuracy. For this purpose, we performed a post-hoc analysis within a cohort of 1431 patients from the COLONPREDICT study, prospectively designed to assess the fecal immunochemical test accuracy in detecting CRC. Over 36.5 ± 8.4 months, cancer was detected in 115 (8%) patients. Patients with CEA values higher than 3 ng/mL revealed an increased risk of cancer (HR 2.0, 95% CI 1.3–3.1), CRC (HR 4.4, 95% CI 1.1–17.7) and non-gastrointestinal cancer (HR 1.7, 95% CI 1.0–2.8). A new malignancy was detected in 51 (3.6%) patients during the first year and three variables were independently associated: anemia (OR 2.8, 95% CI 1.3–5.8), rectal bleeding (OR 0.3, 95% CI 0.1–0.7) and CEA level >3 ng/mL (OR 3.4, 95% CI 1.7–7.1). However, CEA was increased only in 31.8% (95% CI, 16.4–52.7%) and 50% (95% CI, 25.4–74.6%) of patients with and without anemia, respectively, who would be diagnosed with cancer during the first year of follow-up. On the basis of this information, CEA should not be used to assist in the triage of patients presenting with lower bowel symptoms who have recently been ruled out a CRC.

## 1. Introduction

Early colorectal cancer (CRC) diagnosis in primary healthcare is challenging. Most CRCs presenting symptoms, if any, are vague, and often shared among different types of cancer [1]. Furthermore, when most types of cancer develop specific symptoms, they have usually progressed to an advanced stage [2].

The quantitative fecal immunochemical test for hemoglobin (FIT) has shown its usefulness in the diagnosis of CRC both in a screening setting and in the assessment of patients with abdominal symptoms [3,4]. Hence, the National Institute for Health and Care Excellence (NICE) recommends FIT in primary healthcare to assist in the triage of patients presenting with lower bowel symptoms who do not meet the criteria for suspected cancer pathway referral [5].

In addition, carcinoembryonic antigen (CEA), a complex intracellular glycoprotein, is one of the most widely used tumor markers. Serum CEA is produced by approximately 90% of colorectal cancers (CRC), and its most common clinical use is monitoring CRC recurrence following curative resection [6]. However, serum CEA levels can also be elevated in other malignancies and have been proven to be useful in decision-making processes in selected clinical situations unrelated to CRC [7,8,9].

Based on the above, general practitioners sometimes incorporate serum CEA unsupported by evidence as part of health testing for asymptomatic individuals. Despite this, serum CEA is not recommended as a screening test [10]. CEA levels can also be elevated under benign conditions (i.e., cirrhosis, ulcerative colitis), and even smoking appears to almost double the CEA serum concentration in healthy subjects [11]. Moreover, the incidence of other gastrointestinal neoplasia which could also elevate CEA levels is low, and serum CEA lacks sensitivity in early stages [12].

Although CRC is the most common gastrointestinal cancer (and the third most common cancer worldwide) [13], other less prevalent cancer diagnoses have been reported in patients with gastrointestinal symptoms, regardless of their FIT result, as a consequence of the previously commented on lack of specificity of symptoms related to cancer diagnosis [14]. Many of these low prevalence cancers could account for elevated serum CEA levels [7,8,9]. Therefore, the incidence of this group of malignancies in the patient with abdominal symptoms could be sufficiently high to consider assessing the value of CEA in this particular clinical situation.

Therefore, the aim of our study is to evaluate the risk of cancer detection and cancer related death in symptomatic patients that underwent a complete colonoscopy with no CRC according to the serum CEA concentration. We will also evaluate which variables are related to cancer diagnosis in the year after the initial evaluation, as well as the diagnostic accuracy of CEA for cancer detection.

## 2. Materials and Methods

### 2.1. Study Design

This is a post hoc cohort analysis performed within the COLONPREDICT study, which was designed to prospectively evaluate the accuracy of FIT for CRC diagnosis [15]. The study protocol conforms to the ethical guidelines of the 1975 Declaration of Helsinki and was approved by the Clinical Research Ethics Committee of Galicia (Code 2011/038) under a resolution dated 11 April 2012. Patients provided written informed consent prior to inclusion.

We followed the Strengthening the Reporting of Observational studies in Epidemiology statement to conduct and report our study [16].

### 2.2. Study Population

The main characteristics of our cohort have been detailed elsewhere [15]. In short, we included in this analysis ambulatory patients referred consecutively from primary and secondary healthcare for the evaluation of gastrointestinal symptoms. Patients included in the study underwent a colonoscopy, serum CEA and a quantitative FIT. Patients were excluded from this post hoc analysis if a CRC was diagnosed in the baseline colonoscopy or if the colonoscopy was incomplete. We also excluded patients either with insufficient follow-up (less than 2 years) or any untreated cancer diagnosis before basal colonoscopy.

### 2.3. Measurements and Definitions

Serum CEA levels (ng/mL) were measured using a chemiluminescent microparticle immunoassay (UniCel DXI 800; Beckman Coulter, CA, USA). Abnormal levels were defined as levels above 3 ng/mL [15]. Fecal hemoglobin (f-Hb) was measured using OC-SensorTM (Eiken Chemical Co., Tokyo, Japan), as previously reported [17]. Results with f-Hb ≥ 20 µg/g were defined as positive [18].

All the colonoscopies were conducted by endoscopists who perform at least 200 colonoscopies per year [19]. Significant colonic lesion (SCL) was defined as histologically confirmed colitis (any etiology), colonic ulcer, advanced adenoma (any adenoma ≥ 10 mm, with high-grade dysplasia or villous histology), polyposis (>10 polyps of any histology), polyps ≥ 10 mm, bleeding angiodysplasia and complicated diverticular disease (diverticulitis, bleeding). Any diagnosed polyp during baseline colonoscopy was removed either upon that exploration or afterwards.

### 2.4. Follow-Up and Main Outcome

The main outcomes of the study are cancer detection and its related death. Electronic medical records were reviewed for all patients and cancer diagnoses of any etiology were recorded. We classified esophageal, gastric, intestinal, ampullary and colorectal cancer as gastrointestinal cancer (GIC), and we defined an upper GIC as any GIC located outside the colon. The cause and date of death were recorded. We classified the cause of death as (a) global death (death from any cause) and (b) global cancer (death from any cancer). We also divided the cause of death related with cancer into (a) related to GIC, which is further subdivided into (1) upper GIC and (2) CRC, and (b) other types of cancer (non GIC). Secondary outcomes analyzed were cancer risk the first year after performing colonoscopy and the diagnostic accuracy of CEA for cancer detection in this period.

### 2.5. Statistical Analysis

Qualitative variables were expressed as absolute numbers and percentages, while quantitative ones were expressed as medians with their interquartile range. We calculated cumulative risk and number of cases per 1000 patient-years (risk density rate) with their 95% confidence interval (CI) according to the CEA concentration. Patients with normal and abnormal CEA levels were compared using the Fisher’s Exact Test and the Mann Whitney U Test for qualitative and quantitative variables, respectively. We analyzed the differences between both groups in cumulative risk and risk density rate using the Chi-square test and Cochran-Mantel-Haenszel statistics expressed as the risk ratio (RR) and incidence ratio (IR), respectively, with their 95% CI. Cox proportional hazard models were used to estimate the adjusted (age, sex and advanced adenoma) hazard ratios (HR) of presenting a main outcome [20].

A multivariable logistic regression model was used to estimate the independent effect of abnormal serum CEA levels on the detection of any cancer during the first year after baseline colonoscopy, as measured by adjusted odds ratio (OR) with a 95% CI. Variables that had a statistically significant association (*p* < 0.05) with the detection of a new malignancy using Chi-square and Cochran-Mantel-Haenszel statistics were included in the multivariable analysis.

Finally, we assessed the discriminatory ability of serum CEA to detect GIC and non-GIC cancer the first year of follow-up by means of the receiver operating characteristics (ROC) curve and its area under the curve (AUC). Furthermore, we assessed the sensitivity, specificity and positive (PPV) and negative (NPV) predictive value, (positive and negative likelihood ratio and diagnostic odds ratio (DOR) with their 95% CI, using a threshold of 3 ng/mL. Sensitivity analysis was performed using the threshold of 5 ng/mL [21]. Subgroup analysis was conducted to evaluate differences between patients with and without anemia (<11 g hemoglobin per 100 mL in men and <10 g hemoglobin per 100 mL in non-menstruating women), due to its potential association with cancer risk [22,23,24,25]. A *p*-value of <0.05 was deemed statistically significant. Statistical analysis was performed using SPSS statistical software, version 15.0 (SPSS Inc., Chicago, IL, USA).

## 3. Results

### 3.1. Participants

A final sample of 1431 symptomatic patients were included in our analysis (Figure 1).

Of these, 238 (16.6%) had CEA values higher than 3 ng/mL. Patient cohort characteristics are provided in Table 1.

### 3.2. Cancer Diagnosis

During a mean follow-up of 36.5 ± 8.4 months, cancer was detected in 115 (8.0%) patients. Thirty subjects were diagnosed with GIC, and of these 22 (1.5%) lesions were located outside the colon, while 8 (0.6%) were CRC. Furthermore, 85 patients (5.9%) were diagnosed with cancer located outside the gastrointestinal (GI) tract. Of these, twelve patients were diagnosed with lymphoproliferative syndromes (0.8%) and twelve with skin cancer (0.8%), whilst a solid organ neoplasm was diagnosed in fifty-seven patients (4.0%). Four patients showed cancer from an unknown origin.

The distribution of the different types of diagnosed cancer according to the time elapsed since baseline colonoscopy and CEA result is detailed in Appendix A.

### 3.3. Main Outcome

Patients with high CEA values showed an increased risk of death after adjusting for demographic variables and the presence of advanced adenoma (HR 2.6, 95% CI 1.7–3.9). These patients presented both a greater incidence of cancer diagnosis (HR 2.0, 95% CI 1.3–3.1) and cancer-related death (HR 3.7, 95% CI 2.2–6.2) during follow-up. The subgroup analysis also revealed a higher risk for both GIC (HR 2.7, 95% CI 1.3–5.7) and cancer located outside the GI tract (RR 1.7, 95% CI 1.0–2.8). The increased risk of GIC was related to an increased probability of CRC detection (HR 4.4, 95% CI 1.1–17.7) and CRC-related death (HR 8.8, 95% CI 1.6–48.5) in patients with an abnormal CEA concentration. In contrast, the risk of upper GIC and upper GIC-related death did not increase significantly. Table 2 and Figure 2 show the cumulative risk of cancer and related death calculated in Cox’s multivariable regression analysis.

### 3.4. Risk of Cancer Diagnosis during the First Year of Follow-Up

A cancer was detected in 51 (3.6%) patients during the first year after baseline colonoscopy, which represented 44.3% of all diagnosed cancer during follow-up. Nineteen subjects were diagnosed with GIC over this period (63.3% of all diagnosed GIC in this study). Of these, 16 (84.2%) lesions were located outside the colon, while three (15.8%) were CRC. The percentage of CRC and GIC located outside the colon which were diagnosed over this period was 37.5% and 72.7%, respectively. Two thirds of gastroesophageal cancer were diagnosed during the first year. Only three variables were independently associated with cancer detection: rectal bleeding (OR 0.3, 95% CI 0.1–0.7), anemia (OR 2.8, 95% CI 1.3–5.8) and CEA > 3 ng/mL (OR 3.4, 95% CI 1.7–7.1), as shown in Table 3.

### 3.5. Diagnostic Accuracy of CEA for Cancer Diagnosis the First Year of Follow-Up

In our cohort, 652 (45.6%) patients did not have anemia, of whom 96 (14.7%) showed CEA values higher than 3 ng/mL. The AUC of CEA for cancer detection during the first year of follow-up was 0.53 (95% CI, 0.38–0.67) and 0.63 (95% CI, 0.42–0.84) for patients with and without anemia, respectively. At a 3 ng/mL threshold, the sensitivity and specificity for detecting a new cancer in patients without anemia was 50.0% (95% CI, 25.4–74.6%) and 85.9% (95% CI, 83.0–88.4%). For patients with anemia, the sensitivity and specificity for detecting a new cancer in patients was 31.8% (95% CI, 16.4–52.7%) and 83.5% (95% CI, 79.1–87.2%). The diagnostic accuracy of CEA to detect different types of cancer based on the presence of anemia using the thresholds of 3 and 5 ng/mL is shown in Table 4.

## 4. Discussion

### 4.1. Statement of Principal Findings

In this study, we have evaluated the risk of cancer detection in patients with abdominal symptoms and a complete colonoscopy with no baseline CRC detected according to CEA concentration. The risk of cancer, GIC and CRC detection, but not upper-GIC, as well as their related death, is increased in patients with a CEA > 3 ng/mL. Moreover, we have determined three variables independently associated with cancer detection during the first year, as well as the diagnostic accuracy of CEA. In this sense, although specific, CEA values lack sensitivity to identify those patients who would be diagnosed with cancer during the first year of follow-up regardless of the presence of anemia.

### 4.2. Strengths and Weaknesses of Our Study

To our knowledge, this is the first cohort study offering information about the utility of CEA to assist in the triage of patients presenting with lower bowel symptoms who have recently been ruled out a CRC but may be suffering (or not) from other multiple types of cancer due the low specificity of digestive symptoms.

Important strengths of our study are (i) a sample size consisting of a significant number of consecutively recruited patients who underwent a colonoscopy, which guaranteed the absence of CRC and (ii) the availability of a sufficient monitoring period to detect different types of cancer. Both characteristics of our study together guarantee that these results reflect the effectiveness of CEA to predict a cancer diagnosis in the foreseeable future in a patient with abdominal symptoms after ruling out CRC.

Nonetheless, our study is not free from bias. The most important bias is a lack of pertinent background information, such as history of tobacco use or other benign conditions, which could account for a high CEA value without subsequent cancer diagnosis in some patients. This bias may have led to understating the role of CEA as a predictor of cancer. However, it should not have an impact on the assessment of CEA’s sensitivity to detect cancer, only CEA’s specificity.

Moreover, symptomatic patients usually present to primary healthcare, although sometimes they are allowed direct access to the specialist. Our results were obtained from a cohort comprised mainly of secondary healthcare patients. These are generally at a higher risk of being diagnosed with cancer than primary healthcare patients. Consequently, the assessment of CEA’s diagnostic accuracy could be lower in a cohort made up solely of patients from that setting.

### 4.3. Strengths and Weaknesses in Relation to Other Studies

Until now, CEA has been revealed to be mainly related to CRC and our study is along those lines [6]. Even after selecting patients with abdominal symptoms who had undergone a colonoscopy without CRC, a high CEA value only significantly increased the risk of future CRC diagnosis and related death. In this regard, the incidence of CRC during follow-up is very low, within the admissible CRC range values [26].

CEA was previously found to have low sensitivity and specificity to detect CRC in asymptomatic patients. However, it was found that average-risk patients with raised CEA should be investigated, because approximately 9% present some type of cancer. Furthermore, those patients should also be followed up, as another 7% are subsequently diagnosed with some cancer during monitoring [9]. This high cancer incidence could be accounted for the selection process of those patients. Although that study was performed in asymptomatic patients, the reason for which CEA was determined could entail an increased risk of cancer diagnosis. In our prospective cohort, comprised of consecutive patients who complained about abdominal symptoms, 13% of patients were diagnosed with cancer at some time during follow-up. A large proportion of those cancers were detected in the first year of follow-up and were thus likely to be present at the time baseline colonoscopy was performed.

In the past, various CEA thresholds were used. The threshold of 5 ng/mL was the most common during CRC follow-up despite a recommendation to raise this to 10 ng/mL when used in this context as a single test [21]. In this work, we decided to explore a lower threshold, as we are using CEA as a triage test of a group of diseases whose prognosis would benefit from a prompt diagnosis.

### 4.4. Implications for Clinical Practice and Research

The time between the onset of symptoms and consultation with the physician is a significant proportion of the total time to most cancer diagnosis [27]. Unfortunately, the most frequent initial cancer symptoms are common features of benign conditions and are often shared between different types of cancer [28].

Despite previous work recommending a rational evaluation pathway for a patient with a raised CEA [6], our data reveals that CEA also has a poor sensitivity to detect cancer in the patient with abdominal symptoms. The CEA levels were only higher than 3 ng/mL in about one third and one half of patients—with and without anemia, respectively—who will later be diagnosed with cancer during follow-up. Therefore, even as an independent predictor for future cancer diagnosis in the patient with abdominal symptoms who have ruled out a CRC, the determination of serum CEA should not be used as triage tool to identify who could benefit from further diagnostic tests in this population.

### 4.5. Unanswered Questions and Future Research

This study has revealed a remarkably high incidence of non-gastrointestinal cancer in a cohort of patients with abdominal symptoms. Furthermore, the incidence of non-gastrointestinal cancer in the subgroup of patients with CEA below the threshold 3 ng/mL and FIT < 20 µg Hb/g feces exceed the threshold risk of cancer of 3% recommended by the 2015 version of the NICE guidelines for investigation [29]. Similarly, other authors have reported that abdominal symptoms are common at presentation in different types of cancer [30]. Further prospective studies are needed in order to better understand the relationship between the presence of some particular abdominal symptoms and the subsequent diagnosis of an apparently unrelated cancer.

### 4.6. Conclusion

This study assesses the value of CEA in a large cohort of patients with symptoms that could be compatible with multiple tumors after the most common digestive tumor (CRC) had been ruled out with a very reliable test (colonoscopy). In this situation, extremely frequent in clinical practice, some doctors request the determination of serum CEA level to decide whether to perform or not additional work up to assess the possibility of that patient having other types of cancer. Our work, with its limitations, suggests that this practice should be abandoned.

## Figures and Tables

**Figure 1 diagnostics-10-01036-f001:**
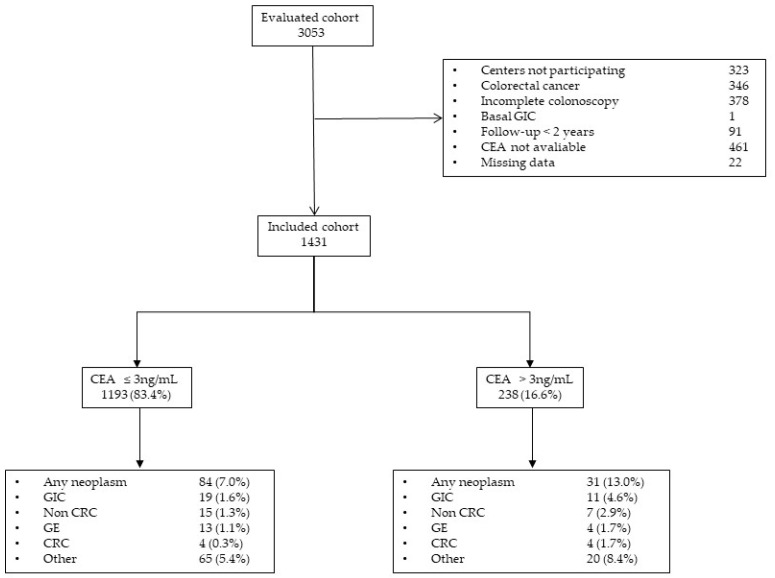
Study population flowchart. CEA = carcinoembryonic antigen; CRC = colorectal cancer; GE = gastroesophageal cancer; GIC = gastrointestinal cancer.

**Figure 2 diagnostics-10-01036-f002:**
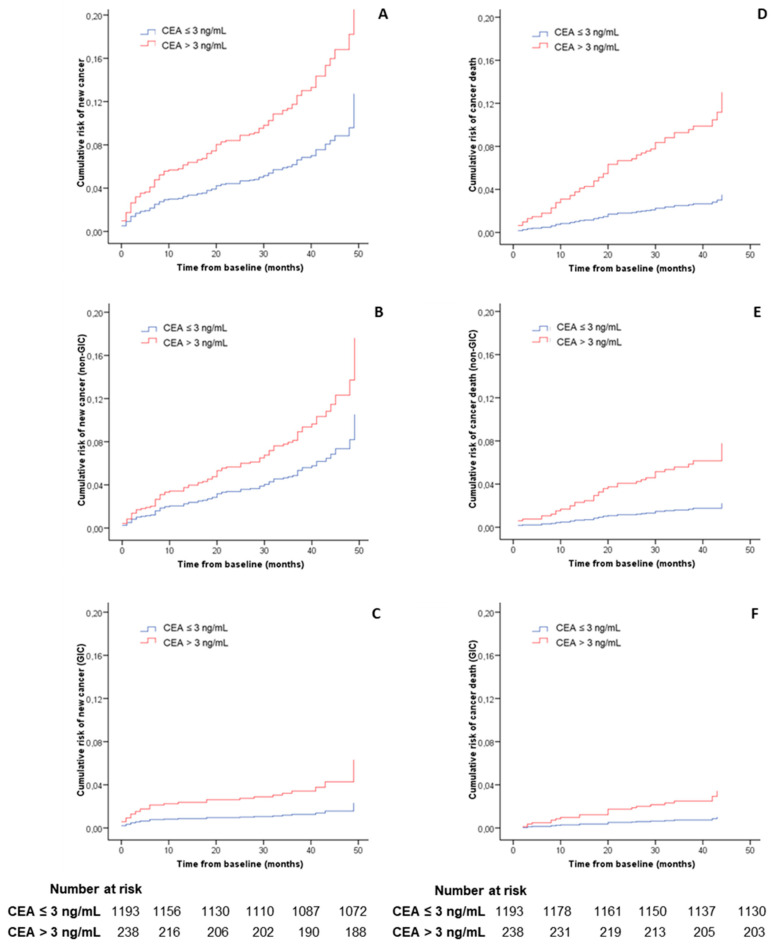
Cumulative risk of cancer and related death. Cumulative risk of cancer and related death during follow-up after baseline evaluation according to carcinoembryonic antigen value and adjusted by sex, age and the presence of advanced adenoma. The figure is calculated with a Cox’s multivariable regression. (**A**) Risk of cancer diagnosis (global cancer); (**B**) risk of non-gastrointestinal cancer diagnosis; (**C**) risk of gastrointestinal cancer diagnosis; (**D**) risk of new cancer-related death; (**E**) risk of new non-gastrointestinal cancer death; (**F**) risk of new gastrointestinal cancer death. Number at risk was calculated for global cancer diagnosis and new cancer related death, respectively.

**Table 1 diagnostics-10-01036-t001:** Characteristics of the individuals included in the analysis.

Characteristics	Overall (*n* = 1431)	CEA ≤ 3ng/mL (*n* = 1193)	CEA > 3 ng/mL (*n* = 238)	*p*
Demographic				
Age, (years)	66.7 (20.0)	66.2 (19.5)	69.1 (22.0)	<0.01
Female sex, no. (%)	748 (52.3)	634 (53.1)	114 (47.9)	0.16
Primary healthcare referral, no. (%)	390 (27.3)	335 (28.1)	55 (23.1)	0.13
Previous colonoscopy, no. (%)	398 (27.8)	326 (27.3)	72 (30.3)	0.38
Daily using ASA, no. (%)	280 (19.6)	223 (18.7)	57 (23.9)	0.07
f-Hb concentration	33.0 (140.0)	33.0 (139.5)	31.0 (142.0)	0.8
Indications, no. (%)				
Rectal bleeding	796 (55.6)	675 (56.6)	121 (50.8)	0.12
Change of bowel habit	839 (58.6)	705 (59.1)	134 (56.3)	0.43
Anaemia ^1^	338 (34.1)	279 (33.4)	59 (38.1)	0.27
Abdominal pain ^1^	451 (45.6)	378 (45.3)	73 (47.1)	0.73
Weight loss ^1^	240 (24.2)	197 (23.6)	43 (27.7)	0.26
Basal colonoscopy findings, no. (%)				
Benign anorectal lesion	612 (42.8)	521 (43.7)	91 (38.2)	0.13
Significant colonic lesions	277 (19.4)	233 (19.5)	44 (18.5)	0.79
Advanced adenoma	206 (14.4)	176 (14.8)	30 (12.6)	0.42
Follow-up, (months)	38.0 (10.0)	38.0 (10.0)	37.0 (11.0)	0.03

^1^ Missing data in 441 subjects; ASA, acetyl salicylic acid; f-Hb, faecal hemoglobin.

**Table 2 diagnostics-10-01036-t002:** Risk of cancer and death according to the carcinoembryonic antigen value.

EVENT	Risk	Overall (*n* = 1431)	CEA ≤ 3 ng/mL (*n* = 1193)	CEA > 3 ng/mL (*n* = 238)	RR/IR (95% CI)	*p*	HR (95% CI)	*p*
Gastrointestinal cancer	Cumulative ^1^	2.1 (1.5–3.0)	1.6 (1.0–2.5)	4.6 (2.6–8.1)	2.9 (1.4–6.0)	<0.01	2.7 (1.3–5.7)	0.01
	Density ^2^	6.9 (4.7–9.9)	5.1 (2.9–8.0)	16.1 (8.0–28.5)	3.1 (1.5–6.5)	<0.01		
	Cumulative death ^1^	1.5 (1.0–2.2)	1.0 (0.6–1.7)	3.8 (2.0–7.0)	3.8 (1.6–8.8)	0.03	3.4 (1.4–8.1)	0.01
	Death density ^2^	4.7 (2.9–7.3)	3.3 (1.8–5.5)	12.8 (5.8–24.4)	4.0 (1.7–9.4)	<0.01		
Upper gastrointestinal cancer ^3^	Cumulative ^1^	1.5 (1.0–2.3)	1.3 (0.8–2.1)	2.9 (1.4–5.9)	2.3 (1.0–5.7)	0.10	2.2 (0.9–5.4)	0.09
	Density ^2^	5.1 (3.3–7.7)	4.0 (2.2–6.6)	10.2 (4.0–20.8)	2.5 (1.0–6.1)	0.04		
	Cumulative death ^1^	1.0 (0.6–1.7)	0.8 (0.5–1.5)	2.1 (0.9–4.8)	2.5 (0.9–7.3)	0.16	2.3 (0.8–6.8)	0.13
	Death density ^2^	3.3 (1.8–5.5)	2.6 (1.5–5.1)	7.3 (2.2–16.8)	2.6 (0.9–7.7)	0.06		
Colorectal cancer	Cumulative ^1^	0.6 (0.3–1.1)	0.3 (0.1–0.9)	1.7 (0.7–4.2)	5.0 (1.3–19.9)	0.04	4.4 (1.1–17.7)	0.04
	Density ^2^	1.8 (0.7–3.7)	1.1 (0.4-2.9)	5.8 (1.5–14.6)	5.3 (1.3–21.3)	<0.01		
	Cumulative death ^1^	0.4 (0.2–0.9)	0.2 (0.0–0.6)	1.7 (0.7–4.2)	10.0 (1.8–54.4)	<0.01	8.8 (1.6–48.5)	<0.01
	Death density ^2^	1.5 (0.4–2.9)	0.4 (0.0–1.8)	5.8 (1.5–14.6)	10.6 (1.9–57.8)	<0.01		
Non gastrointestinal cancer	Cumulative ^1^	5.9 (4.8–7.3)	5.4 (4.3–5.5)	8.4 (5.5–12.6)	1.5 (1.0–2.5)	0.11	1.7 (1.0–2.8)	0.04
	Density ^2^	19.7 (15.7–24.5)	17.9 (13.9–22.6)	29.6 (17.9–45.7)	1.7 (1.0–2.7)	0.05		
	Cumulative death ^1^	2.7 (1.9–3.6)	1.8 (1.2–2.8)	6.7 (4.2–10.6)	3.6 (1.9–6.8)	< 0.01	3.5 (1.8–6.7)	<0.01
	Death density ^2^	8.8 (6.2–11.7)	5.8 (3.7–9.1)	22.6 (13.1–36.9)	3.8 (2.0–7.3)	<0.01		
Cancer	Cumulative ^1^	8.0 (6.7–9.6)	7.0 (5.7–8.6)	13.0 (9.3–18.0)	1.8 (1.3–2.7)	<0.01	2.0 (1.3–3.1)	<0.01
	Density ^2^	27.0 (22.3–32.1)	23.4 (18.6–28.9)	46.8 (31.8–66.5)	2.0 (1.3–3.0)	<0.01		
	Cumulative death ^1^	4.1 (3.2–5.3)	2.9 (2.0–4.0)	10.5 (7.2–15.0)	3.7 (2.2–6.1)	<0.01	3.7 (2.2–6.2)	<0.01
	Death density ^2^	13.5 (10.2–17.2)	9.1 (6.2–12.8)	35.8 (23.0–52.6)	3.9 (2.3–6.5)	<0.01		
Death	Cumulative ^1^	7.0 (5.8–8.4)	5.4 (4.2–6.8)	15.1 (11.1–20.2)	2.8 (1.9–4.1)	<0.01	2.6 (1.7–3.9)	<0.01
	Density ^2^	22.6 (18.6–27.4)	17.2 (13.1–21.9)	51.5 (35.8–71.2)	3.0 (2.0–4.5)	<0.01		

^1^ Cumulative risk is expressed as percentage and its 95% CI; ^2^ risk density rate is expressed per 1000 patient-years and its 95% CI; ^3^ defined as a cancer located outside the colon. CI = confidence interval; HR = adjusted hazard ratio; IR = incidence ratio; RR = risk ratio.

**Table 3 diagnostics-10-01036-t003:** Factors associated with cancer detection the first year after baseline colonoscopy.

	New Cancer	Odds Ratio (95% CI)	Odds Ratio (95% CI) ^1^
Sex			
Male (683)	28 (4.0%)	1	
Female (748)	23 (3.1%)	0.8 (0.4–1.3)	
Age			
≤70 years (846)	19(2.2%)	1	
>70 years (585)	32(5.5%)	2.4 (1.4–4.3)	
Primary healthcare referral			
No (1041)	35 (3.4%)	1	
Yes (390)	16 (4.1%)	1.2 (0.7–2.2)	
Rectal bleeding			
No (635)	37 (5.8%)	1	1
Yes (796)	14 (1.8%)	0.3 (0.2–0.6)	0.3 (0.1–0.7)
Change of bowel habit			
No (592)	25 (4.2%)	1	
Yes (839)	26 (3.1%)	0.7 (0.4–1.3)	
Anaemia			
No (652)	12 (1.8%)	1	1
Yes (338)	22 (6.5%)	3.5 (1.8–7.1)	2.8 (1.3–5.8)
Abdominal pain			
No (539)	21 (3.9%)	1	
Yes (451)	13 (2.9%)	0.7 (0.4–1.5)	
Weight loss			
No (750)	25 (3.3%)	1	
Yes (240)	9 (3.4%)	1.1 (0.5–2.4)	
Faecal immunochemical test			
≤10 µg/g (*n* = 948)	30 (3.2%)	1	
>10 µg/g (*n* = 483)	21 (4.3%)	1.4 (0.8–2.4)	
Carcinoembryonic antigen value			
≤3 ng/mL (*n* = 1193)	31 (2.3%)	1	1
>3 ng/mL (*n* = 238)	20(8.4%)	3.2 (1.9–5.6)	3.4 (1.7–7.1)
Advanced adenoma			
No (1225)	43 (3.5%)	1	
Yes (206)	8 (3.9%)	1.1 (0.5–2.3)	

^1^ Variables with statistically significant differences were introduced in a multivariable logistic regression analysis stepwise (backward) to identify the prognostic factors associated with cancer detection the first year after baseline colonoscopy; as a result of this method, the dichotomous variable “Age” has finally not entered in the final equation. CI = confidence interval.

**Table 4 diagnostics-10-01036-t004:** Diagnostic accuracy of carcinoembryonic antigen for different types of cancer diagnosed in the first year during follow-up.

Threshold	Type of Cancer	Anaemia	Prevalence	%AT	Sensitivity ^1^	Specificity ^1^	NPV ^‡,1^	PPV ^1^	AUC	LR+	LR−	DOR
CEA > 3 ng/mL	GI cancer	No	0.6	14.7	50.0 (15.0–85.0)	85.5 (82.6–88.0)	99.6 (98.7–99.9)	2.1 (0.6–7.3)	0.70 (0.41–0.99)	3.4	0.6	5.82
		Yes	3.0	17.5	30.0 (10.8–60.3)	82.9 (78.5–86.6)	97.5 (94.9–98.8)	5.1 (1.7–13.9)	0.57 (0.39–0.74)	1.8	0.8	2.08
	Upper GI cancer ^2^	No	0.2	14.7	0.0 (0.0–79.3)	85.3 (82.3–87.8)	99.8 (99.0–99.96)	0.0 (0.0–3.8)	0.48 (0.44–0.53)	0.0	NA	NA
		Yes	3.0	17.5	30.0 (10.8–60.3)	82.9 (78.5–86.6)	97.5 (94.9–98.8)	5.1 (1.7–13.9)	0.57 (0.39–0.74)	1.8	0.8	2.08
	Colorectal cancer	No	0.5	14.7	66.7 (20.8–93.9)	85.5 (82.6–88.0)	99.8 (99.0–99.96)	2.1 (0.6–7.3)	0.77 (0.42–1.0)	4.6	0.4	11.80
		Yes	0.0	17.5	NA	NA	NA	NA	NA	NA	NA	NA
	Non-GI cancer	No	1.2	14.7	50.0 (21.5–78.5)	85.7 (82.8–88.5)	99.3 (98.2–99.7)	4.2 (1.6–10.2)	0.60 (0.32–0.87)	3.5	0.6	6.00
		Yes	3.6	17.5	33.3 (13.8–60.9)	83.1 (78.7–86.8)	97.1 (94.4–98.5)	6.8 (2.7–16.2)	0.49 (0.28–0.70)	2.0	0.8	2.46
	Cancer	No	1.8	14.7	50.0 (25.4–74.6)	85.9 (83.0–88.4)	98.9 (97.7–99.5)	6.3 (2.9–13.0)	0.63 (0.42–0.84)	3.6	0.6	6.11
		Yes	6.5	17.5	31.8 (16.4–52.7)	83.5 (79.1–87.2)	94.6 (91.3–96.7)	11.9 (5.9–22.5)	0.53 (0.38–0.67)	1.9	0.8	2.37
CEA >5 ng/mL	GI cancer	No	0.6	5.1	50.0 (15.0–85.0)	99.2 (93.3–96.6)	99.7 (98.8–99.9)	6.1 (1.7–19.6)	0.70 (0.41–0.99)	10.5	0.5	19.91
		Yes	3.0	8.6	20.0 (5.7–51.0)	91.8 (88.3–94.3)	97.4 (95.0–98.7)	6.9 (1.9–22.0)	0.57 (0.39–0.74)	2.4	0.9	2.79
	Upper GI cancer ^2^	No	0.2	5.1	0.0 (0.0–79.3)	94.9 (93.0–96.4)	99.8 (99.1–99.97)	0.0 (0.0–10.4)	0.48 (0.44–0.53)	0.0	NA	NA
		Yes	3.0	8.6	20.0 (5.7–51.0)	91.8 (88.3–94.3)	97.4 (95.0–98.7)	6.9 (1.9–22.0)	0.57 (0.39–0.74)	2.4	0.9	2.79
	Colorectal cancer	No	0.5	5.1	66.7 (20.8–93.9)	95.2 (93.3–96.6)	99.8 (99.1–99.97)	6.1 (1.7–19.6)	0.77 (0.42–1.0)	14.0	0.4	39.88
		Yes	0.0	8.6	NA	NA	NA	NA	NA	NA	NA	NA
	Non-GI cancer	No	1.2	5.1	37.5 (13.7–69.4)	95.3 (93.4–96.7)	99.2 (98.1–99.7)	9.1 (3.1–23.6)	0.60 (0.32–0.87)	8.1	0.7	12.27
		Yes	3.6	8.6	16.7 (4.7–44.8)	91.7 (88.2–94.2)	96.8 (94.1–98.2)	6.9 (1.9–22.0)	0.49 (0.28–0.70)	2.0	0.9	2.21
	Cancer	No	1.8	5.1	41.7 (19.3–68.0)	95.6 (93.8–97.0)	98.9 (97.7–99.5)	15.2 (6.7–30.9)	0.63 (0.42–0.84)	9.5	0.6	15.61
		Yes	6.5	8.6	18.2 (7.3–38.5)	92.1 (88.6–94.6)	94.2 (91.0–96.3)	13.8 (5.5–30.6)	0.53 (0.38–0.67)	2.3	0.9	2.59

AUC = area under the curve; CEA = carcinoembryonic antigen; DOR = diagnostic odds ratio; GI = gastrointestinal; %AT = percentage of carcinoembryonic antigen above threshold; LR = likelihood ratio; NA = not applicable; NPV = negative predictive value; PPV = positive predictive value; +: positive; −: negative. ^1^ Values were expressed as percentages and their 95% confidence interval. ^2^ Defined as a cancer located outside the colon. ‡ Some NPV results were rounded to two decimals, as they could be incorrectly interpreted if they were rounded to one decimal (100.0).

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
