# Peer review of "Predictive Value of Carcinoembryonic Antigen in Symptomatic Patients without Colorectal Cancer: A Post-Hoc Analysis within the COLONPREDICT Cohort"

_diagnostics, 2020, doi:10.3390/diagnostics10121036_

Round 1

Reviewer 1 Report

Dear Editor,

Thank you for giving me this opportunity to review this manuscript. In this manuscript, the authors did a posthoc analysis of the COLONPREDICT study to examine the predictive value of carcinoembryonic antigen (CEA) in predicting new gastrointestinal (GI) cancer in patients free of colorectal cancer (CRC), which were documented colonoscopically. From my point of view, there are some concerns about this article.

  1. The main problem of this article is what kind of medical care these patients received during the post-colonoscopic follow-up period. These patients did not routinely receive a 2nd-time colonoscopic exam during the follow-up period. Patients with a higher CEA may be more alert and then received a more intensive follow-up. A higher CRC incidence during the 3-year period in the “CEA>3 group” may just reflect a more intensive medical care. Those who did not diagnose with a new CRC did not mean that he/she has no new cancer.
  2. Besides, we can see that over the average 3-year follow-up period, there were only 4 new CRC cases diagnosed in each group (4/1,139 vs. 4/238). The positive predictive value is relatively low.
  3. CEA is also secreted by lung cancer, and many studies had pointed out blood CEA may serve as a predictor or disease activity indicator for lung cancer. Did these patients receive any chest exam? Patients without CRC but have a higher CEA may have undiagnosed lung cancer or upper GI cancer.

minor:

  1. On page 10, line 28, “Uther” is a typo.

Author Response

  • The main problem of this article is what kind of medical care these patients received during the post-colonoscopic follow-up period. These patients did not routinely receive a 2nd-time colonoscopic exam during the follow-up period. Patients with a higher CEA may be more alert and then received a more intensive follow-up. A higher CRC incidence during the 3-year period in the “CEA>3 group” may just reflect a more intensive medical care. Those who did not diagnose with a new CRC did not mean that he/she has no new cancer.
  • Besides, we can see that over the average 3-year follow-up period, there were only 4 new CRC cases diagnosed in each group (4/1,139 vs. 4/238). The positive predictive value is relatively low.
  • CEA is also secreted by lung cancer, and many studies had pointed out blood CEA may serve as a predictor or disease activity indicator for lung cancer. Did these patients receive any chest exam? Patients without CRC but have a higher CEA may have undiagnosed lung cancer or upper GI cancer.

Response: I would like to answer all three previous points with only one response as, to my opinion, all of them are related.

We agree with Reviewer 1 that the methodology of this work has significant biases, but we also believe that those do not invalidate the main conclusions of our study.

This is a retrospective study where data about the follow up of the COLONPREDICT cohort were collected through the review of the patients ‘medical records: No patient was scheduled for any specific diagnostic work up, that is, there was no follow-up protocol, but cancers diagnosed over the following years after baseline colonoscopy (including described lung cancers) were detected during the “real clinical practice” by their physicians (outside any research protocol).

For this reason, the real cumulative incidence of any type of cancer during the following years since baseline colonoscopy, may be higher than the one described in our manuscript for any subgroup of patients in our cohort regardless CEA level (on the assumption that there could be incident cancers which still weren´t diagnosed at the end of the follow up). However, the duration of the follow-up is long enough to allow any prevalent cancer (at the time of baseline colonoscopy and CEA determination) to produce complications which would lead to its diagnosis regardless of its stage at the beginning of the follow up.

All this, together with the possibility that at least some of the patients  could have benign conditions not recorded in our database which may justify an increase in CEA levels, in example chronic kidney injury or active smoking, may have resulted in understating the risk of cancer in the subgroup of patients with  CEA values above the threshold in our study.

Despite this limitation of our study, commented in the discussion section of the manuscript, the main conclusions of an ulterior hypothetical prospective study are likely to be similar: The presence of serum CEA levels above 3 ng / ml is an independent risk factor for many types of cancer diagnosis (maybe even more important than what has been quantified in our study). However, the incidence of those types of cancer is too low to justify an ulterior screening which should not be limited to one only type of diagnostic test due to the variety of lesions detected during the follow-up of this study.

Furthermore, the global incidence of cancer in patients with digestive symptoms who present CEA levels below the chosen threshold is too high to limit testing to those subjects with abnormal levels of this tumor marker. Our data analysis suggest that determination of serum CEA should not be used as a triage test in a patient with abdominal symptoms with a low risk of colorectal cancer to decide whether to perform or not additional work up to assess the possibility of that patient having other type of cancer.

  • On page 10, line 28, “Uther” is a typo.

Response: corrected.

Thank you very much for the time spent reviewing our work and your constructive criticism.

Reviewer 2 Report

The authorsof the manuscript "Predictive Value of Carcinoembryonic Antigen in Symptomatic Patients without Colorectal Cancer. Post-hoc Analysis within COLONPREDICT Cohort" wanted to assess the risk of cancer in patients  (N=1431) with abdominal symptoms after a complete colonoscopy without colorectal cancer (CRC), primarily based on the value of carcinoembryonic antigen (CEA), with 3 ng/mL being the border between "low" and "high" value. Some other parameters, such as fecal hemoglobin were also determined.

This study is correctly performed, but does not bring anything new in the field.

Introductory part is not sufficiently informative. The authors should offer some more relevant literature data. For example, when claiming "to evaluate differences between patients with and without anemia due to its potential association with cancer risk", the reader would like to see a valid reference.

The authors should introduce abbreviations where needed (ROC, receiver operating characteristics) and introduce the full name, for some terms (PPV, positive predictive value). The list with abbreviations would be welcome (for example, the authors introduce "G" for gastroesophageal cancer only in the figure legend 1, not earlier). They should very clearly explain the difference between "other cancer" and "global cancer" (categories under "d" and "e").

The statistical methods were performed correctly. As a suggestion: Optimal way to analyze data presented in Table 1 would be Hotelling's T2 test or, even better, two-group multivariate permutation test. These are very simple and very powerful multivariate statistical test. However, there is no need for re-analyzing these data because all relevant p-values are higher than 0.05. I do not believe that changing the the test applied would change the final result of analyses.

Addition of the number of patients at risk at the x- axis in Fig. 2, would be useful. The set of pictures should be properly labeled: A, B, C, D, E, F, and properly explained in the figure legend. I wonder why the authors did not determine the concentration of CEA during the follow-up, maybe annually.

The results deserve to be discussed in a broader terms. The authors should have made functional links with data presented on Figure 2 and discuss relevant findings which can be find in the current biomedical literature in the field.  I also wonder why the authors did not cite some relevant papers showing the false positive rate of CEA with respect to diseases relevant for this paper.

Finally, I appreciate tremendous work dedicated in preparing all tables, and especially the one presented as the supplementary one.

Thank you.

Author Response

  • This study is correctly performed, but does not bring anything new in the field.

Response: This study does not offer a "positive" result (CEA has not shown usefulness in the studied situation) but points out a practical conclusion which had not previously been explored.

To our knowledge, this is the first cohort study offering information about the utility of CEA to assist in the triage of patients presenting with lower bowel symptoms who have recently been ruled out a CRC but may be suffering (or not) from  other multiple types of cancer due the low specificity of digestive symptoms.

Until now, the existing literature has assessed the value of CEA in decision-making procedures about many particular situations, (for example, to treat or not a pancreatic cyst, assessing the possibility of pancreatic cancer as a target) and there is vast information on the relationship between CEA and colorectal cancer covering every phase of its management: a) CEA value in CRC screening setting, b) CRC diagnosis or c) in the prognosis of diagnosed CRC.

However, there was not a “global approach” of the utility of CEA, assessing the value of this biomarker in a large cohort of patients with symptoms that could be compatible with multiple tumors after the most common digestive tumor (CRC) had been ruled out with a very reliable test (colonoscopy). This study sets out from a “general situation” – the uncertainty of existing digestive symptoms and a colonoscopy without CRC- and considers any type of tumor as a target during a long monitoring period.

In this “general situation”, extremely frequent in clinical practice, some doctors request the determination of a tumor marker as well-known as CEA, unsupported by evidence. Our work with its limitations suggests that this practice should be abandoned. The use of CEA as a screening tool (patient with no symptoms) has been already investigated and is discouraged, but not as a triage tool (in the assessment of symptoms).

  • Introductory part is not sufficiently informative. The authors should offer some more relevant literature data. For example, when claiming “to evaluate differences between patients with and without anemia due to its potential association with cancer risk”, the reader would like to see a valid reference.

Response: we have added some references where this issue has been assessed (line 153).

  • The authors should introduce abbreviations where needed (ROC, receiver operating characteristics) and introduce the full name, for some terms (PPV, positive predictive value). The list with abbreviations would be welcome (for example, the authors introduce "G" for gastroesophageal cancer only in the figure legend 1, not earlier). They should very clearly explain the difference between "other cancer" and "global cancer" (categories under "d" and "e").

Response: we have added aforenamed abbreviations at lines 146-149. “G” abbreviation was replaced by “GE” at line 160 which is the one used in its corresponding Figure 1. We tried to clarify the different types of cancer in lines 124-127

  • The statistical methods were performed correctly. As a suggestion: Optimal way to analyze data presented in Table 1 would be Hotelling's T2 test or, even better, two-group multivariate permutation test. These are very simple and very powerful multivariate statistical test. However, there is no need for re-analyzing these data because all relevant p-values are higher than 0.05. I do not believe that changing the the test applied would change the final result of analyses.

Response: thank you very much for this information. It will be very useful in future studies.

  • Addition of the number of patients at risk at the x- axis in Fig. 2, would be useful. The set of pictures should be properly labeled: A, B, C, D, E, F, and properly explained in the figure legend. I wonder why the authors did not determine the concentration of CEA during the follow-up, maybe annually.

Response: Figure 2 was corrected. We did not determine the concentration of CEA during the follow up because COLONPREDICT investigators did not monitor this cohort. Clinical decisions about any treatment or diagnostic test after baseline colonoscopy were made by their physicians out from any research protocol.

  • The results deserve to be discussed in a broader terms. The authors should have made functional links with data presented on Figure 2 and discuss relevant findings which can be find in the current biomedical literature in the field. I also wonder why the authors did not cite some relevant papers showing the false positive rate of CEA with respect to diseases relevant for this paper.

Response: We have tried to interpret the results presented in tables and figures as functionally as possible. Moreover, we have summarized them in a specific conclusion for clinical practice. However, we are open to add in the discussion section additional information on the differences between our results with respect to those from any other published work considered relevant at the discretion of both the reviewers and the editors of the journal.

The target of our research is broader than others which addressed more specific situations. Thus, some hypothetical false positives of other studies focused, for example, on pancreatic cancer as a target, could not be so when other types of cancer are taken into account as a target. The bias caused by the lack of information on benign situations which can justify increases in CEA level has already commented in the discussion section, and may be relevant  when  quantifying the risk of having a CEA value above the threshold but is not likely to invalidate the most important conclusion of this manuscript, which is CEA has low sensitivity not only to detect colorectal cancer in screening setting but also for detecting cancer in patients with digestive symptoms and a normal colonoscopy.

  • Finally, I appreciate tremendous work dedicated in preparing all tables, and especially the one presented as the supplementary one.

Response: We deeply thank the reviewer for this comment

Thank you very much for the time spent reviewing our work and your suggestions.

Round 2

Reviewer 1 Report

Although this study does not offer a "positive" result, I suggest that the authors present their novel "practical conclusion" in the abstract and the conclusion sections, which are lacking in the present form.

The authors have pointed out their "practical conclusion" in response to reviewer 2's comments as below. Please integrate the following contents and make it more clear for the readers to understand. Some English grammar needs to be checked.

This study sets out from a “general situation” – the uncertainty of existing digestive symptoms and a colonoscopy without CRC- and considers any type of tumor as a target during a long monitoring period. In this “general situation,” extremely frequent in clinical practice, some doctors request the determination of a tumor marker as well-known as CEA, unsupported by evidence. Our work with its limitations suggests that this practice should be abandoned. The use of CEA as a screening tool(patient with no symptoms) has been already investigated and is discouraged, but not as a triage tool (in the assessment of symptoms).

Author Response

Dear reviewer,

We edited the discusion section of the manuscript to add a small conclusion.

Thank you very much again for the time spent in your comprehensive peer review.

Sincerelly,

Noel Pin Vieito

Reviewer 2 Report

Dear Authors,

Thank you for the revised, improved version of your manuscript. 

All the best,

Author Response

Sincerelly,

Noel Pin Vieito